# Bioprocesses with Reduced Ecological Footprint by Marine *Debaryomyces hansenii* Strain for Potential Applications in Circular Economy

**DOI:** 10.3390/jof7121028

**Published:** 2021-11-30

**Authors:** Silvia Donzella, Claudia Capusoni, Luisa Pellegrino, Concetta Compagno

**Affiliations:** Department of Food, Environmental and Nutritional Sciences (DeFENS), University of Milan, Via L. Mangiagalli 25, 20133 Milan, Italy; silvia.donzella@unimi.it (S.D.); claudia.capusoni@unimi.it (C.C.); luisa.pellegrino@unimi.it (L.P.)

**Keywords:** *Debaryomyces hansenii*, marine yeast, yeast proteins, yeast lipids, phytase, yeast cultivation, circular economy

## Abstract

The possibility to perform bioprocesses with reduced ecological footprint to produce natural compounds and catalyzers of industrial interest is pushing the research for salt tolerant microorganisms able to grow on seawater-based media and able to use a wide range of nutrients coming from waste. In this study we focused our attention on a *Debaryomyces hansenii* marine strain (Mo40). We optimized cultivation in a bioreactor at low pH on seawater-based media containing a mixture of sugars (glucose and xylose) and urea. Under these conditions the strain exhibited high growth rate and biomass yield. In addition, we characterized potential applications of this yeast biomass in food/feed industry. We show that Mo40 can produce a biomass containing 45% proteins and 20% lipids. This strain is also able to degrade phytic acid by a cell-bound phytase activity. These features represent an appealing starting point for obtaining *D. hansenii* biomass in a cheap and environmentally friendly way, and for potential use as an additive or to replace unsustainable ingredients in the feed or food industries, as this species is included in the QPS EFSA list (Quality Presumption as Safe—European Food Safety Authority).

## 1. Introduction

In recent years, the importance of developing sustainable processes has been increasing in parallel with the concern regarding global climate change. In this context, microbial fermentations can offer several advantages compared with other systems for the possibility to convert waste into high value products. This technology represents a big opportunity to promote a sustainable economy, and optimization of industrial bioprocesses is very relevant [1]. An efficient cellular performance requires the cultivation of selected microorganisms under conditions suitable for optimal growth and metabolism. In contrast, biotechnological processes are often performed under conditions that can cause cell stress, such as the presence of high osmotic pressure due to high concentration of nutrients, or the presence of inhibitors [2]. Submerged fermentations as well as bioconversions largely consume freshwater. A challenge to perform biotechnological processes, saving this precious resource, can be the use of seawater-based media. It was calculated that the water footprint of bioethanol ranges from 1388 to 9812 litres of water for each litre of ethanol produced. Gerbens-Leenes and Hoekstra [3] concluded that due to the concern about global freshwater shortages, the water usage issue could soon be included in the food and land usage debate [3,4]. Thus, the use of seawater for preparing fermentation media could be an attractive approach for some biotechnological productions. Indeed, some fermentation processes in seawater-based media to produce proteins, secondary metabolites, and other biomolecules are already covered by patents [5,6]. Additionally, seawater contains a spectrum of minerals that can enrich essential nutrients fermentation media. On the basis of these considerations, marine yeasts, possessing several unique characteristics for higher osmo-tolerance compared to their terrestrial counterparts, represent a very useful source of microbial agents. This is also true of their enzymes, which present features often more suitable for developing bioconversions. For these reasons, marine yeasts have great potential to be applied in industrial biotechnological processes [7,8,9,10]. 

*D. hansenii* is a yeast species known for halotolerant traits [11], and, being included in the QPS EFSA list (Quality Presumption as Safe—European Food Safety Authority), can be appealing for feed or food-related applications [12,13]. Strains isolated from cheese and fish intestine have recently been investigated for potential probiotic properties [14]. Other strains have been investigated for usage in biocontrol of ochratoxigenic moulds [15], for production of enzymes like exopeptidases and thermophilic β-glucosidases, and for production of xylitol and riboflavin (Vitamin B2) [16]. *D. hansenii* relevance is linked also to applications for contributing to final aroma and composition in food [17]. However, *D. hansenii*’s biotechnological potential was suffering because of scarce availability of studies about cultivation in a bioreactor under conditions suitable for industrial bioprocesses. Studies have been in fact carried out mainly for xylitol production due to the biotechnological importance of this compound [18]. Tavares et al. reported on the growth in continuous culture on a chemically defined medium at controlled pH 5.5 for production of xylitol [19]. Studies on genes involved in xylose transport and xylitol production have been reported [20].

Recently, we investigated [21] osmotic stress response in *D. hansenii* strains isolated from gasteropod gill (*Ifremeria nautilei*) and from coral [22]. In the present study, we developed processes for cultivation of the marine strain Mo40, evaluating the use of conditions suitable for freshwater-saving processes and waste recycle. Cultivations were performed in a bioreactor on seawater-based media and containing as carbon source a mixture of glucose and xylose, simulating lignocellulosic hydrolysates. In addition, urea was utilized as a cheap nitrogen source. In the aim to optimize conditions for industrial bioprocesses, the influence of low pH on fermentation parameters was analysed. We investigated the potential use of this yeast as a microbial cells factory, for production of natural products like single cell proteins (SCP) as well as of single cell oils (SCO) of industrial interest. Phytase activity was also tested. Phytases are enzymes that catalyse the release of phosphate from phytic acid and are commonly exploited as additives to reduce phytic acid content in feed/food. 

## 2. Materials and Methods

### 2.1. Yeast Strain 

The yeast strain used in this work was *D. hansenii* Mo40 [22]. For long-term storage, the yeast strain was maintained at −80 °C on 15% (*v/v*) glycerol and 85% (*v/v*) YPD medium.

### 2.2. Media Composition

YPD: yeast extract 10 g/L, peptone 20 g/L and glucose 20 g/L.

MMP: mineral medium was employed for pre-inoculum and for fermentation in the bioreactor. Medium composition as reported in [23] with some modifications: glucose 20 g/L, (NH_4_)_2_SO_4_ 5 g/L, MgSO_4_* 7 H_2_O 0.5 g/L, KH_2_PO_4_ 3 g/L, trace metals (disodic EDTA 15 mg/L, ZnSO_4_* 4 H_2_O 4.5 mg/L, MnCl* 4 H_2_O 0.1 mg/L, CoCl_2_* 6 H_2_O 0.3 mg/L, CuSO_4_* 5 H_2_O 0.3 mg/L, Na_2_MoO_4_* 2 H_2_O 0.4 mg/L, CaCl_2_* 2 H_2_O 4.5 mg/L, FeSO_4_ * 7 H_2_O 3 mg/L, H_3_BO_3_ 1 mg/L, KI 0.1 g/L) and vitamins (D-biotin 0.05 mg/L, Ca D(+)panthotenate 1 mg/L, nicotinic acid 1 mg/L, myoinositol 25 mg/L, thiamine hydrochloride 1 mg/L, pyridoxol hydrochloride, p-aminobenzoic acid 0.2 mg/L).

MMPSS: sea salts at 40 g/L (S9883 Sigma-Aldrich, St. Louis, MO, USA) were added to MMP.

MMPhy: glucose 20 g/L, (NH_4_)_2_SO_4_ 5 g/L, MgSO_4_* 7H_2_O 0.5 g/L, phytic acid 0.11 g/L sodium salt hydrate (68388 Sigma-Aldrich, USA), trace metals and vitamins as in MMP. The pH was adjusted at pH 4.5 with the addition of H_2_SO_4_.

IMSS (Industrial-like medium): glucose 33 g/L, xylose 16 g/L, sea salt 40 g/L, yeast extract 2 g/L, corn steep solid (C8160 Sigma-Aldrich, USA) 5 g/L, urea 2 g/L, KH_2_PO_4_ 3 g/L, H_2_SO_4_ 2 mL/L, CaCl_2_ 0.1 g/L, NaCl 0.1 g/L.

Medium B: glucose 50 g/L, (NH_4_)_2_SO_4_ 1 g/L, KH_2_PO_4_ 1 g/L, MgSO_4_ * 7H_2_O 0.05 g/L, NaCl 0.01 g/L, CaCl_2_ 0.01 g/L, yeast extract 1 g/L; 0.1 M MES (2-(N-morpholino) ethanesulfonic acid) buffer/KOH (to maintain pH 6).

“Lapeña” medium [24]: glucose 20 g/L, peptone 30 g/L, yeast extract 20 g/L.

### 2.3. Fermentation Conditions

Yeast cells were pre-cultured on MMP medium (pH 6 maintained by addition of MES buffer/KOH). Cultivations were run at 28 °C in a rotary shake at 150 rpm in bluffed flasks for 24 hours, cells were harvested by centrifugation at 5000 rpm and washed three times with sterile NaCl solution (9 g/L). They were then used to inoculate the bioreactor at initial OD_600nm_ 0.05. 

Aerobic batch cultivations were performed in an Applikon system bioreactor with a working volume of 1 L. The temperature was set at 28 °C, the stirring speed at 500 rpm, and the pH, measured by Applisens pH electrode, was adjusted and maintained at 6 or 4.5 by automatic addition of 5 M KOH or 2 M H_2_SO_4_. The dissolved oxygen concentration (maintaining more than 30% of air saturation) was measured by an Applisens polarographic oxygen probe. 

For lipid production, cultivations were carried out in 500 mL bluffed flask with 100 mL of Medium B. In this case, pre-inoculum was performed in YPD medium and cells were inoculated at an initial concentration of 0.1 OD_600nm_ from seed cultures. 

### 2.4. Quantification of Biomass and Compounds

Samples were harvested from the bioreactor at appropriate intervals and used to monitor the cell growth measuring the optical density at 600 nm with a spectrophotometer, after appropriate dilution. 

For dry weight determination, washed culture samples were filtered on a 0.45 µm glass microfiber GF/A filter (Whatman, Maidstone, UK) and dried 24 h at 80 °C. 

The concentration of carbon sources, such as glucose and xylose on supernatants, was determined by commercial enzymatic kits (Roche, cat. numb. 1 0716251 035, Basel, Switzerland and Megazyme cat number K-XYLOSE 04/18, Bray, Ireland respectively). All the assays were performed in triplicate and the standard deviations varied between 1 and 5%. Specific consumption rates of glucose and xylose were calculated during the exponential phase of growth. The yield of biomass was calculated as the total amount of dry weight divided by the consumed carbon sources.

Total nitrogen concentration in culture supernatants was determined by the Kjeldahl method using a SpeedDigester K-376 and a KjelMaster K-375 (Buchi Italia, Cornaredo, Italy). 

The content of total amino acids (except tryptophan) was determined on the liofilized samples. Acid hydrolysis and oxidation of cyst(e)ine and methionine to cysteic acid and methionine sulphone, respectively, were carried out according to EC regulation No 152/2009 [25]. The hydrolysate was adjusted to pH 2.2 and analyzed with a Biochrom 30plus amino acid analyzer (Biochrom, Cambridge, UK) adopting the conditions described [26]. 

Lipid content was determined via the sulpho-phospho-vanilline colorimetric method (Spinreact, Girona, Spain) on washed cell pellets (30 OD_600nm_), suspended in 0.5 mL of cold redistilled water. The assays were performed in triplicate and standard deviations varied between 1 and 5%. Lipid yield is reported as g lipids/g d.w.

### 2.5. Phytase Sequence

To isolate genomic DNA, pellets of 30 OD_600nm_ of cells were resuspended in 0.5 ml of 0.05 M Tris–HCl/0.02 M EDTA at pH 7.5. This suspension was transferred to a precooled tube with an equal volume of glass beads (425–600 µm). Mechanical lysis was performed using a TissueLyser LT alternating 2 min of agitation at 50 Hertz with 1 min in ice for 4 cycles. The supernatant was added with 25 μL of SDS 20% (*w/v*) and incubated at 65 °C for 30 min. Immediately, 0.2 mL of 5 M potassium acetate was added and the tubes were placed on ice for 30 min. Samples were centrifuged at 13,000 rpm for 5 min and supernatant was transferred to a fresh microcentrifuge tube. The DNA was precipitated by adding 1 volume of isopropanol. After incubation at room temperature for 5 min, the tubes were centrifuged for 10 min. The DNA was washed with 70% ethanol and dissolved in 50 µL of TE RNAsi (10 mMTris-HCl, 1 mM EDTA, pH 7.5 RNAsi 100 µg/mL). Samples were incubated at 37 °C for 30 min [27].

Phytase sequences of *D. hansenii* Mo40 were obtained in this work. The phytase gene was amplified from gDNA using primers: Forward Phy1 CCG ACC ATG GAT GGT ATC GAT TTC C, Reverse Phy2 CAT CGG ATC CTA ATT GTC ACC GGA. Primers were designed employing *D. hansenii* CBS 767 sequence (GeneID: 2900382). PCR amplification was carried out by denaturing at 98 °C for 7 min, followed by 30 cycles of denaturing at 98 °C for 10 s, annealing at 59 °C for 30 s, extension at 72 °C for 45 s, and a final extension at 72 °C for 10 min. The produced amplicon was cloned in a plasmid psf URA TPI and sequenced by Microsynth AG company (Balgach, Switzerland). Mo40 and CBS767 phytase sequences were aligned and compared using http://multalin.toulouse.inra.fr/multalin/online (accessed on 2 November 2021) tools.

### 2.6. Determination of Phytase Activity

Cells were pre-cultured for 24 h in YPD, harvested by centrifugation at 5000 rpm, and washed three times with sterile NaCl solution (9 g/L) and inoculated at initial OD_600nm_ 1 in MMPhy. After 48 h of incubation, phytase activity was determined.

Extracellular enzymatic activity was detected on the supernatant and cell-bound (intracellular) activity using whole cells. The activity was measured by orthophosphate production, following the ammonium molibdate blue method as reported in [28] with some modifications. For extracellular activity determination, cell cultures were centrifuged at 13,000 rpm, and 1 mL of supernatant was added to 4 mL buffer composed of 0.2 M Na acetate/acetic acid and 8 mM phytic acid at pH 4.5. To determine cell-bound activity we used a homogeneous cells suspension. We set up enzymatic activity using a standard amount of cells 50 OD_600nm_ (corresponding to almost 10 mg d.w. depending on the strain) in a final volume of 5 mL. Cells were collected and washed twice with 0.2 M Na acetate/acetic acid pH 4.5 and resuspended in a final volume of 1 mL. Cell suspension was added to 4 mL of buffer 0.2 M Na acetate/acetic acid, 8 mM phytic acid at pH 4.5. All buffers employed to test enzymatic activity were prewarmed at reaction temperature. Blank was assembled using 1 ml of 0.2 M Na acetate/acetic acid at pH 4.5 and 4 mL of 0.2 M Na acetate, 8 mM phytic acid and treated as sample.

For enzymatic activity determination, a 5 mL reaction was incubated in 15 mL tube at 37 °C and stirred at 300 rpm. The reaction was immediately stopped (time 0) and stopped after 15, 30, 60, and 120 min. Reaction was stopped by mixing 0.5 mL of reaction with 0.5 mL TCA 5% solution, samples were centrifuged 3 min at 13,000 rpm, and the supernatant was collected. In order to determine orthophosphate concentration, 0.4 mL of supernatant was added to 0.4 mL of molibdate solution. This solution was prepared daily, by mixing solution A and B in a ratio of 4:1 (solution A: 2.6% N_6_H_24_Mo_7_O_24_* 4H_2_O and 5.5% H_2_SO_4_; solution B: 4.6% FeSO_4_*7H_2_O). The sample was incubated 10 min at 25 °C and read against blank at OD_700nm_. Phosphate concentration was determined using a standard curve for KH_2_PO_4_. A unit of phytase is defined as the amount of protein that hydrolyzes 1 µmol phosphorus per min. Specific activity is expressed in mU/mg of cell dry weight. To determine the effect of temperature, samples prepared with prewarmed buffer were incubated at 50 °C and 60 °C.

## 3. Results and Discussion

### 3.1. Cultivation in Bioreactor: Optimization of Growth Conditions

With the aim to optimize cultural conditions suitable for industrial bioprocesses, we analyzed the growth of the *D. hansenii* Mo40 strain in a bioreactor. In a previous work, we studied the effects caused by the cultivation in presence of sea salts in two *D. hansenii* strains isolated from the marine environment [19]. Based on our results, we decided to proceed with our attention on the Mo40 strain, which showed the best performance even in presence of high salt concentration (2 M NaCl). Simulating the use of seawater-based media, we performed cultures on media containing sea salts (MMPSS medium). For comparison, cultures were arranged also on media without sea salt additions (MMP medium, control cultures). In addition, we evaluated the effect of sea salts on growth parameters at low pH. Cultivation at low pH is preferred for industrial processes because it reduces risks of bacterial contaminations, that are more common at neutral pH. For this purpose, cultures were run at two different pH values, 4.5 and 6, in a bioreactor under controlled aerobic conditions, and the initial pH values were maintained all along the process. 

Figure 1 shows the kinetics of growth obtained under different conditions. The absence of any lag phase on media containing sea salts suggested that the marine *D. hansenii* strain quickly adapted to their presence, being the pre-inoculum cultured on medium without sea salts. A high degree of adaptation under different growth conditions is considered a relevant trait for industrial strains, reducing the time of the process. By comparing the results obtained from cultures at pH 6, we can deduce that the presence of sea salts did not cause significant differences on growth parameters (Table 1), except for glucose consumption rate, which increased. The same is also true if we compare MMP cultures (control cultures) performed at pH 6 and pH 4.5 (Table 1). In this case it is possible to notice that no significantly negative effects on growth rate resulted from lowering the pH of the process. An insignificant lower biomass yield and a slightly lower glucose consumption rate were observed at pH 4.5. In contrast, the presence of sea salts (MMPSS medium) produced differences in all the parameters of cultures performed at pH 4.5. In particular, in the processes carried out at this pH, we observed a reduction of 6% in the growth rate, of 10% in the biomass yield, and of 13% in the glucose consumption rate, in comparison to the process run at pH 6 (Table 1). These results suggest that at lower pH the presence of sea salts in the medium causes a higher metabolic burden, with some negative consequences on growth performance. A strong negative effect on growth was reported to be caused by high pH combined with salt (NaCl) [29]. In contrast, the scale-up in the bioreactor had a strong positive effect on the process performance in terms of biomass yield, in comparison with the cultures performed in shaken flasks [21]. Growth rates and glucose consumption rates are in fact similar to the ones previously reported [21], but biomass yields showed an increase of 36%, from 0.38 g_d.w._/g_c.s._ detected in flask fermentations to an average of 0.58 g_d.w._/g_c.s._ detected in the bioreactor system (Table 1). This improvement indicates that oxygen limitation, occurring in the flask, exerts a strong negative effect on biomass yield, which can be fully overcome in a bioreactor, where it is possible to maintain the dissolved oxygen concentration over 30% of air saturation. The more efficient aeration system in a bioreactor plays a very important role for reaching high biomass concentration of *D. hansenii,* which it is known to exhibit a respiratory metabolism [21].

In conclusion, we need to consider that the reductions observed in the presence of sea salts at low pH, although moderate, can be compensated for by the possibility of using conditions that are more advantageous for run bioprocesses. The capability to grow and reach high biomass yield under these conditions is essential because it reduces the use of freshwater and reduces contaminations, keeping in mind that high osmotic pressure and acidic pH are conditions not suitable for the cultivation of most microorganisms.

### 3.2. Simulation of an Industrial Bioprocess with Reduced Ecological Footprint

In order to set up a process by simulating industrial conditions with reduced ecological footprint, cultivations were performed on a seawater-based medium containing glucose and xylose as carbon sources, a sugar composition similar to a lignocellulosic hydrolisate (IMSS). Urea and corn steep were employed as nitrogen sources. This aspect is also important, because these materials are cheap and widely present in industrial media formulation [30]. These processes were run in a bioreactor under controlled aerobic conditions and at pH 4.5. 

Analysis of the growth kinetic reported in Figure 2 leads to the conclusion that *D. hansenii* is able to efficiently grow under these conditions. As expected, this yeast was able to consume glucose and xylose [18], and in particular, it is interesting to point out that in *D. hansenii* no glucose repression was observed, being as both carbon sources simultaneously metabolized (Figure 2). This is an appreciated trait in industrial bioprocesses employing agri-food waste because it reduces the production time and, consequently, the costs. This kind of material readily contains mixtures of sugars and other compounds, and the capability to consume all of them at the same time provides advantages in terms of productivity.

The results reported in Table 1 indicate that IMSS composition did not negatively affect *D. hansenii* fitness. The principal growth parameters, such as growth rate, glucose consumption rate, and biomass yield remained similar in fact to those observed in the process carried out on mineral medium (MMPSS, pH 4.5). In addition, the utilization of urea as nitrogen source, instead of ammonium salts, was likewise efficient. At the end of the process, a biomass concentration of 28.5 g/L was reached, with a biomass yield of 0.6, indicating that this medium is promising for *D. hansenii* industrial cultivation.

In conclusion, the ability to grow on seawater-based media and at low pH coupled with the use of a wide range of carbon and nitrogen sources makes *D. hansenii* a good candidate to be employed for developing industrial bioprocesses with a low ecological footprint. This exerts a positive impact on costs and on the environment, addressing one of the main goals of the circular economy, which is the reduction of waste by recycling.

### 3.3. SCP and SCO Production

The growing world population is forcing the search for alternative nutritional sources that can replace or supplement those currently used. This is true also in the case of animal nutrition, in order to sustain intensive breeding. Yeasts are among the preferred candidates, due to their content of high-value compounds, such as protein, lipids, and vitamins. Studies have illustrated that several yeast proteins exhibit favourable amino acid composition, and yeast biomass can provide vitamins (mainly the B group), showing excellent properties in animal diets, giving immunological and health benefits [31,32]. Lipids, especially TAGs, are accumulated over 20% and up to 70% of their dry cell mass by oleaginous microorganisms that possess this ability. In addition to applications for biofuel production [33], microbial lipids can be considered as components for animal and human diets because they have a composition very similar to vegetable oils [34,35]. There is a strong interest now by industries in finding alternative ways to produced oils and fatty acids (e.g., polyunsaturated fatty acids—PUFAs) to be used as feed/food additives [36]. Oleaginous yeasts belong to the genera *Lipomyces, Yarrowia, Cryptococcus, Trichosporon*, and *Rhodosporidium* [37], but *D. hansenii* was reported also as a lipid accumulating species [18]. 

Because few data about these skills are available in literature for *D. hansenii*, we decided to test both SCP and SCO production in the marine strain Mo40. To assess protein content and amino acid composition, we utilized biomass generated by batch fermentations using IMSS medium. The protein content was found to be approximately 23% of dry weight. In comparison to values reported in other species [24], this level is lower. However, the amino acid composition was very similar to that reported by Lapeña et al. [24] (Table 2). To understand if the lower protein content was due to the cultivation medium used, which was very different from the one reported by these authors, cells were cultivated in the same medium [24]. We found that the protein content increased to 45%, indicating that medium composition deeply affects the protein content of yeast biomass. By using a rich medium, the protein content of *D. hansenii* Mo40 was comparable to well established meat alternatives such as Quorn by *Fusarium venenatum* [38] and oncom by *Neurospora intermedia* [39].

In order to evaluate the ability to produce SCO, *D. hansenii* Mo40 cells were cultured on lipidogenic medium characterized by low nitrogen content, known to induce lipids accumulation [27]. Under these conditions (Figure 3), after 24 h of cultivation, due to nitrogen exhaustion, lipids started to accumulate and after 72 h of cultivation the biomass reached a lipid content of 20.7%. Other oleaginous yeast species produce lipids at higher amounts but, due to the QPS status of *D. hansenii*, this can represent an interesting starting point to improve this process. 

In conclusion, we demonstrate the appealing possibility to obtain *D. hansenii* biomass with good protein and lipid contents, by means of the fermentation processes.

### 3.4. Phytase Activity

Recently, marine yeasts have been explored as source of enzymes useful in bioprocesses [7,8,9,10]. The salt tolerance showed by these enzymes makes them more resistant to industrial bioprocess conditions [40]. Phytate-degrading activities represent a class of industrially relevant enzymes for the feed and fuel industries [41]. Because monogastric animals are not able to metabolize phytate, their feeds are often fortified with inorganic phosphorus, thus increasing the final cost. In addition, phytic acid has negative effects on health, as it is a chelating agent that reduces bioavailability of proteins and ions [42]. At present, attention about this anti-nutritional compound is increasing also for humans. Phytate degradation in food is mediated mainly by fermentations processes led by phytate-degrading microorganisms [43] or during food processing by the endogenous phytases present in the food matrix [44]. 

*D. hansenii* is able to grow using phytic acid as a sole phosphorus source [45]. To characterize phytase activity in the Mo40 strain, we started sequencing the gene encoding in order to compare a marine phytase sequence with its terrestrial counterpart (reference strain CBS 767, isolated from beer [46]). The alignment of the two sequences shows some differences. In particular, the Mo40 gene presents 46 nucleotidic substitutions that result in nine different amino acids (Figure 4). The presence of the conserved motifs RHGERYP (72-79 aa) and HD (334-335 aa) confirms that this enzyme belongs to the phytase acid class. Furthermore, the absence of a secretion signal sequence makes it possible to speculate that *D. hansenii* phytase could be a cell-bound enzyme. To assess phytase activity, *D. hansenii* Mo40 was cultivated in MMPhy medium, with phytic acid as the sole phosphorus source. Phytase activity was then assayed on cells as well as on supernatants, at pH 4.5 and at two temperatures, 37 °C and 60 °C. The lack of activity in the supernatant confirmed that phytase in *D. hansenii* is a cell-bound enzyme, with specific activity corresponding to 0.57 mU/mg_d.w._ and 5.03 mU/mg_d.w._, respectively, detected at 37 °C and at 60 °C (Table 3). Resilience at high temperature can be requested at the industrial level because heat treatments are commonly adopted to limit spoilage and during pelleting processes in feed manufacture [40].

In conclusion, Mo40 phytase could be relevant to increase bioavailability of protein and mineral content in feed/food. In addition, this activity can reduce environmental pollution due on the excretion of undigested phytate. In this respect, it could be interesting to evaluate the capability to degrade phytic acid also in seawater, because yeast biomasses can be added to aquaculture feed as sources of nutrients. 

## 4. Conclusions

In this work we studied the optimization of bioprocesses with reduced ecological footprint for potential applications in the circular economy by using a marine strain of *D. hansenii.* We show that this strain is able to grow at low pH and with high biomass yield on seawater-based media containing glucose/xylose mixtures and urea as cheap nutrients. Preliminary characterizations indicate that this yeast strain can produce SCP and SCO and degrade phytic acid. The above capabilities coupled with its inclusion in the QPS EFSA list make *D. hansenii* a good candidate to be used in several food/feed industrial fields. By combining these characteristics, it would be possible to produce natural compounds and relevant enzymes by means of seawater-based fermentations coupled with a wide range of cheap carbon and nitrogen sources obtained from waste. This can improve the overall economics of the processes and have a positive impact not only on saving freshwater for other uses, but also on the fulfilment of one of the main challenges of circular economy, which is the reduction of waste by recycle.

## Figures and Tables

**Figure 1 jof-07-01028-f001:**
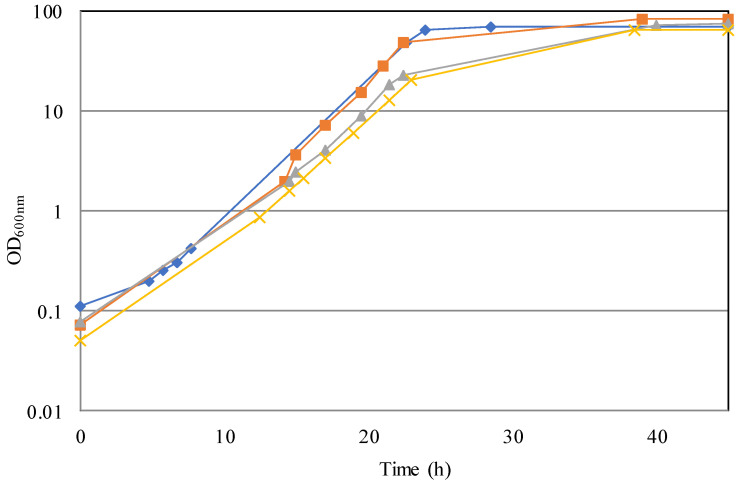
Growth of *D. hansenii* Mo40 strain in mineral media at different pHs without and with sea salts (SS). Blue: MMP pH 4.5, orange: MMPSS pH 4.5, grey: MMP pH 6, yellow: MMPSS pH 6.

**Figure 2 jof-07-01028-f002:**
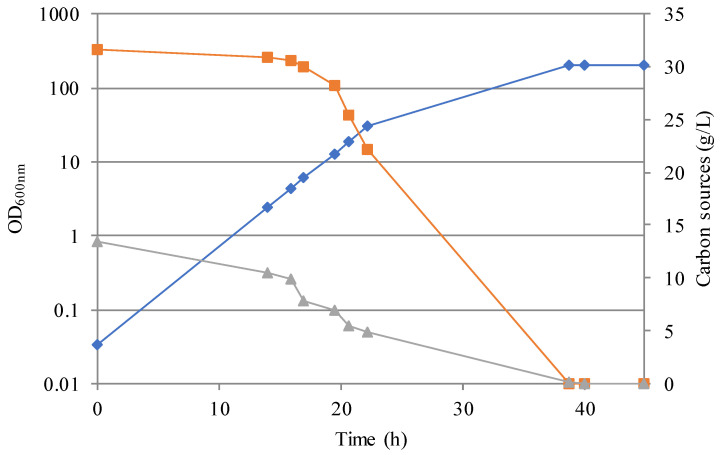
Growth of *D. hansenii* in industrial-like seawater-based medium (IMSS). Blue: OD_600nm_; orange: residual glucose; grey: residual xylose.

**Figure 3 jof-07-01028-f003:**
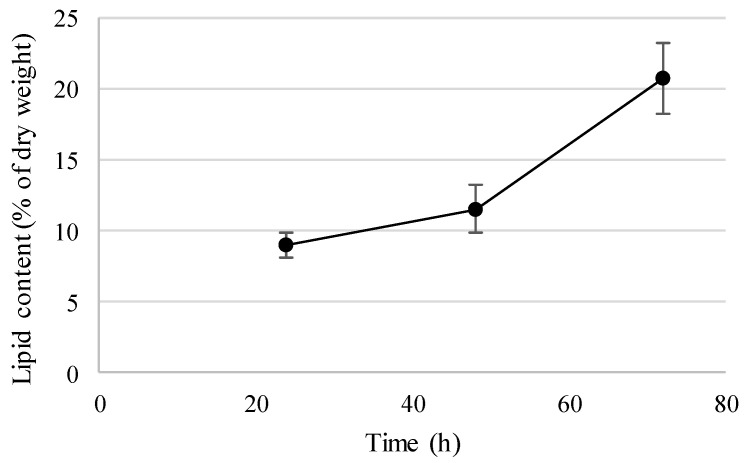
Lipid production by the Mo40 strain cultivated in lipidogenic medium (high C/N ratio).

**Figure 4 jof-07-01028-f004:**
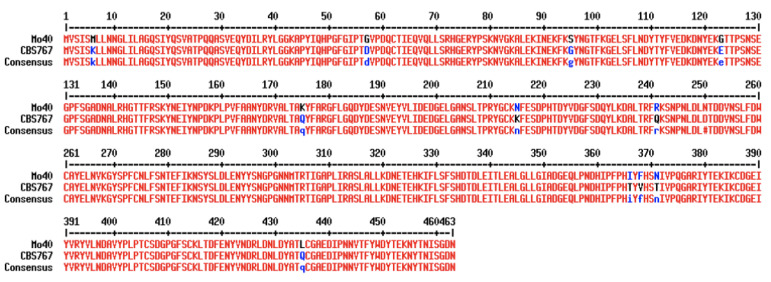
Alignment between phytase sequences of Mo40 (marine strain) and CBS 767 (*D. hansenii* type strain). The different amino acids are indicated in blue, the conserved ones in red.

**Table 1 jof-07-01028-t001:** Growth parameters of *D. hansenii* Mo40 cultivated in the absence (MMP) or presence of sea salts (MMPSS) and in an industrial-like seawater-based medium (IMSS).

		µ max	Final Dry Weight	Biomass Yield	q Glucose	q Xylose
		[h^−1^]	[g/L]	[g_d.w._/g_c.s._]	[mmol_glc_/g_dw_/h]	[mmol_xiyl_/g_dw_/h]
pH 4.5	MMP	0.32 ± 0.010	13.5 ± 0.5	0.61 ± 0.016	2.83 ± 0.098	-
	MMPSS	0.30 ± 0.008	12.3 ± 0.6	0.55 ± 0.015	2.47 ± 0.088	-
	IMSS	0.30 ± 0.011	28.5 ± 0.9	0.62 ± 0.018	2.56 ± 0.144	2.30 ± 0.100
pH 6	MMP	0.31 ± 0.012	13.8 ± 0.4	0.63 ± 0.012	2.57 ± 0.121	-
	MMPSS	0.34 ± 0.013	13.4 ± 0.5	0.62 ± 0.015	3.25 ± 0.132	-

c.s.: consumed sugar.

**Table 2 jof-07-01028-t002:** Amino acid composition of Mo40 biomass obtained after growth in industrial-like seawater-based medium containing glucose and xylose (IMSS).

	mg/g_DW_	g/100 g Protein	mg/g_DW_ (g/100 g Protein) Reported by Lapeña et al. [22]
Asp	20	10.2	37–48 (7–10)
Thr	14	5.9	21–26 (4–5)
Ser	13	6.4	21–28 (4–6)
Glu	27	13.2	64–76 (13–15)
Gly	9.6	4.8	19–30 (4–6)
Ala	12	5.9	23–29 (5–6)
Val	13	5.7	20–29 (4–6)
Cys ^a^	5	2.5	3–6 (0.6–1.2)
Met ^b^	2.5	1.1	4–7 (0.8–1.4)
Ile	11	4.7	16–24 (3–5)
Leu	18	8.1	29–38 (6–8)
Tyr	6	3.0	11–18 (2–4)
Phe	10	4.6	14–19 (3–4)
Lys	16	7.2	27–43 (5–9)
His	4.5	2.0	9–13 (2–3)
Arg	10	4.4	21–32 (4–6)
Pro	8	4.0	17–23 (3–5)
Total		94.7	

^a^: cysteic acid; ^b^: methionine sulphone.

**Table 3 jof-07-01028-t003:** Phytase activity (mU/mg_d.w_) detected in cells grown in the presence of phytic acid as the sole phosphorus source. Enzyme activity was assayed at pH 4.5.

	Cell-Bound	Extracellular
60 °C	5.03 ± 0.513	BDL
37 °C	0.57 ± 0.071	BDL

BDL: below detection limit.

## Data Availability

The datasets used and/or analysed during the current study are available from the corresponding author on reasonable request.

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
