# Peer review of "Bioprocesses with Reduced Ecological Footprint by Marine Debaryomyces hansenii Strain for Potential Applications in Circular Economy"

_jof, 2021, doi:10.3390/jof7121028_

Round 1

Reviewer 1 Report

Single cell protein, single cell oil and phytase produced by different species of marine yeasts have been investigated by many researchers. Many marine yeasts can produce much more SCP, SCO and have higher phytase activity than D. hansenii Mo40 used in this study. So this study is not new and meaningless. In addition, the title is not suitable. There are to many mistakes in English grammar and description in the text. In fact, it is very difficult to use seawater as the medium for cultivation of marine yeasts because the bioreactor can be easily destroyed by seawater.

Author Response

We thank the Reviewer for the comments, the text has been improved and mistakes corrected.

By looking at literature and comparing with our results, among 327 yeast strains isolated from seawater and other marine environments and tested for crude protein content by Chi et al (2008), the best results were obtained by Cryptococcus aureus Z114 (41%).

Yarrowia lipolytica SWJ-1b isolated from the marine fish gut and its mutant contained from 47.6% to 53.7% (w/w) crude protein on a maximum of 20.8 g/l of biomass (Cui et al., 2011), values comparable to the level found in our D.hansenii Mo40. Up to date, the commercialized products containing Single Cell Proteins are produced employing terrestrial strains (Weibe et al. 2017).

With regard to phytase enzymes from marine microorganisms, the only relevant activity was found in Kodamea ohmeri BG3 isolated from the gut of a marine fish, that can reach over 557.9 mU/ml of phytase activity (Li et al., 2009).

The majority of bioreactors used in labs are in stainless steel 316L (Applikon.com), a marine grade steel with enhanced corrosion-resistance properties that make it ideal for use in marine industry (Material Properties Data: Marine Grade Stainless Steel). At industrial scale corrosion could be avoided by applying a suitable coating layer inside the bioreactor, and considering the use of pipes made from Chlorinated Polyvinyl Chloride (CPVC) or Polyvinyl Chloride (PVC), as reported by Zaky et al. 2018.

Reviewer 2 Report

I found the manuscript to be very interesting and relevant in using sea water as an alternative. We do not always consider the scarcity of fresh water in fermentation. Overall there is enough "novelty" to be considered for acceptance but there are however quite a lot of error that need to be addressed before that.

One "scientific" issue I have is the inclusion of the phytase-encoding gene. It does not really add to the story so I would recommend removing it.

Line 52: safety typo

Line 54: "gut" could use a less crass word

line 55: have typo

Line 66: Debaryomyces typo

Line 74: as a micriobial

Line 76: clumsy sentence: should maybe not in past tense

Line 80: only one strain was used in this study

Line 87: not sure * is the way to denote the bound water

Line 90 d-biotin and d-panthotenate. the d should be capitalized and two fontsizes smaller

Line 97: what does "simil" mean

Line 98: where does the corn steep solid come from?

Line 107: three typo

Line 179: buffers

Line 207: "in fact" should be omitted, "because allows to reduce" -- "as it reduces"

Line 214, 216: try to avoid the use of "very"

Line 253, Figure 1: the colours do not match up.

Line 274, Figure 2: Again the colours do not match up

Line 293: the word "high-quality" seems inappropriate

Line 298: unique seems inappropriate

Line 309: found is a clumsy word here, weight typo

Line 312: wondered is not a scientific term

Line 313: hypothesis typo

Line 314: increased is better  than "raised"

Line 315: deeply typo

Line 338: thus is better than "so"

Line 344: grow typo, omit "by"

Reference are not all according to the MDPI guidelines  example Line 420

Figure 4: not sure what is meant with the consensus when alignments do not match

Author Response

We thank the Reviewer for the overall positive evaluation of the Manuscript. The text has been checked and corrected according to reviewers’ suggestions.

One "scientific" issue I have is the inclusion of the phytase-encoding gene. It does not really add to the story so I would recommend removing it.

Being commonly exploited as additive to reduce phytic acid content in feed/food, we find that the phytase activity produced by D.hansenii Mo40 strain could be interesting for industrial applications, because D. hansenii is a GRAS organism. The high cell-bound activity at high temperature (65°C) could be in fact required in feed production processes during pelleting and heat treatment (Mrudula Vasudevan et al., 2019).

Line 52: safety typo

Line 54: "gut" could use a less crass word

line 55: have typo

Line 66: Debaryomyces typo

Line 74: as a micriobial

Line 76: clumsy sentence: should maybe not in past tense

Line 80: only one strain was used in this study

Line 87: not sure * is the way to denote the bound water

Line 90 d-biotin and d-panthotenate. the d should be capitalized and two fontsizes smaller

Line 97: what does "simil" mean

Line 98: where does the corn steep solid come from?

Line 107: three typo

Line 179: buffers

Line 207: "in fact" should be omitted, "because allows to reduce" -- "as it reduces"

Line 214, 216: try to avoid the use of "very"

Line 253, Figure 1: the colours do not match up.

Line 274, Figure 2: Again the colours do not match up

Line 293: the word "high-quality" seems inappropriate

Line 298: unique seems inappropriate

Line 309: found is a clumsy word here, weight typo

Line 312: wondered is not a scientific term

Line 313: hypothesis typo

Line 314: increased is better  than "raised"

Line 315: deeply typo

Line 338: thus is better than "so"

Line 344: grow typo, omit "by"

We are grateful to the Reviewer for his/her suggestion. The text has been changed following his/her comments.

Reference are not all according to the MDPI guidelines example Line 420  

Figure 4: not sure what is meant with the consensus when alignments do not match

The online tool MultAlin (http://multalin.toulouse.inra.fr/multalin/) has been used to compare Mo40 and CBS767 phytase sequences. The sequence alignment output is displayed as a coloured image containing the consensus sequence, where a residue that is highly conserved appears in red and as an uppercase letter in the consensus line. A residue that is weakly conserved appears in blue and as a lowercase letter in the consensus line. In the caption we referred just to red/blue colors because, when the comparison is between two sequences as in this case, the consensus line is less informative but still included in the output. We maintained the conservation thresholds and other alignment parameters at the values suggested by MultAlin.

Reviewer 3 Report

Review for the paper

Bioprocesses with Reduced Ecological Footprint by Marine Debaryomyces hansenii Strain for Potential Applications in Circular Economy

submitted in JoF

Debaryomyces hansenii has a long history with humans as a cheese ripening culture

this species is then included in QPS EFSA list (Quality Presumption as Safe - European Food Seafty Autority).

Debaryomyces hansenii marine strain (Mo40)

what is the genetic background of this strain, compared to strains already used in cheese making

full genome published, comparative genomics? metabolomics?

Authors optimized cultivation in bioreactor at low pH on seawater-based media containing mixture of sugars (glucose and xylose) and urea. Under these conditions the strain exhibited high growth rate and biomass yield.

Mo40 can produce a biomass containing 45% proteins and 20%  lipids.

Compared to fungal biomass such as Quorn?   ascomycete fungus Fusarium venenatum

Protein from renewable resources: Mycoprotein production from agricultural residues

Open Access

Upcraft, T., Tu, W.-C., Johnson, R., (...), Hallett, J., Guo, M.       2021        Green Chemistry

23(14), pp. 5150-5165

The Biotechnology of Quorn Mycoprotein: Past, Present and Future Challenges Whittaker, J.A., Johnson, R.I., Finnigan, T.J.A., Avery, S.V., Dyer, P.S.     2020        Grand Challenges in Biology and Biotechnology

  1. 59-79

OR  filamentous fungus Neurospora intermedia

Fungi burger from stale bread? a case study on perceptions of a novel protein-rich food product made from an edible fungus

Open Access

Hellwig, C., Gmoser, R., Lundin, M., Taherzadeh, M.J., Rousta, K.          2020        Foods

9(8),9081112

etc etc

very active research field

This strain is also able to degrade phytic acid by a cell-bound phytase activity.

Phytase activity  was also tested. Phytases are enzymes that catalyse the release of phosphate from phytic  acid and are commonly exploited as additive to reduce phytic acid content in feed/food.

very interesting aspect

it would be possible to produce natural compounds and relevant enzymes by means of seawater-based fermentations

please provide current industrial productions based on seawater fermentation

coupled with a wide range  of cheap carbon and nitrogen sources obtained from wastes.

please provide current industrial productions based on seawater fermentation coupled with a wide range  of cheap carbon and nitrogen sources obtained from wastes.

or even coupled with a wide range  of cheap carbon and nitrogen sources obtained from wastes, without seawater.

please enhance the quality of the captions

example

Figure 1. Growth of D. hansenii Mo40 strain under different conditions. SS indicates presence of sea salts. Blu: MMP pH  4.5, red: MMPSS pH 4.5, green: MMP pH 6, purple: MMPSS pH 6.

Blue not Blu

all details should be provided in the caption

so the reader is able to understand the figure without going throughout the text

IMSS (simil-Industrial Medium): glucose 33 g/ L, xylose 16 g/ L, sea salt 40 g / L, yeast  extract 2 g/ L, corn steep solid 5 g/ L, urea 2 g/ L, KH2PO4 3 g / L, H2SO4 2 mL/ L, CaCl2 0.1  g/ L, NaCl 0.1 g/ L.

quite expensive medium with yeast extract…

 comparison with

MARINE BROTH contains all the nutrients necessary to cultivate the majority of marine bacteria, lacking the agar usual solidifier component. Our Marine Broth is prepared according to ZoBell, containing almost double the mineral content of sea water. The high salt content helps to simulate sea water.

or ZoBell??

Author Response

Debaryomyces hansenii has a long history with humans as a cheese ripening culture

 this species is then included in QPS EFSA list (Quality Presumption as Safe - European Food Seafty Autority).

 Debaryomyces hansenii marine strain (Mo40)

what is the genetic background of this strain, compared to strains already used in cheese making

full genome published, comparative genomics? metabolomics?

Mo40 strain used in this study was isolated in 2009 by Burgaud et al. from a deep-sea coral (-2300m). By the sequencing of the D1/D2 domain of the 26S rRNA, Mo40 was identified as D. hansenii with 100% similarity with the type strain, and it is now available as UBOCC-A-208035 at the UBO Culture Collection. To date sequence of the full genome and other -omics studies are not available.

 Authors optimized cultivation in bioreactor at low pH on seawater-based media containing mixture of sugars (glucose and xylose) and urea. Under these conditions the strain exhibited high growth rate and biomass yield.

 Mo40 can produce a biomass containing 45% proteins and 20% lipids.

 Compared to fungal biomass such as Quorn?   ascomycete fungus Fusarium venenatum

 Protein from renewable resources: Mycoprotein production from agricultural residues

Open Access

Upcraft, T., Tu, W.-C., Johnson, R., (...), Hallett, J., Guo, M.       2021        Green Chemistry

23(14), pp. 5150-5165 The Biotechnology of Quorn Mycoprotein: Past, Present and Future Challenges Whittaker, J.A., Johnson, R.I., Finnigan, T.J.A., Avery, S.V., Dyer, P.S.     2020        Grand Challenges in Biology and Biotechnology

  1. 59-79

The dry mass of F. venenatum contains 48% proteins and 12% fat, quantities not so different to the ones obtained from D.hansenii Mo40. For Quorn production, the fungus is cultivated in a synthetic medium with glucose, ammonium and supplemented with biotin. The costs associated with the substrate results in a high price for SCP (Souza Filho et al., 2018). In our opinion, the rapid growth of D.hansenii and the capability of growing in seawater-based media contribute to making it a competitive hosts for SCP production. A sentence for comparison with our results and the reference have been added now in the revised version of the manuscript.

OR  filamentous fungus Neurospora intermedia

 Fungi burger from stale bread? a case study on perceptions of a novel protein-rich food product made from an edible fungus

Open Access

Hellwig, C., Gmoser, R., Lundin, M., Taherzadeh, M.J., Rousta, K.          2020        Foods

9(8),9081112

 etc etc

We found that the best process with N. intermedia is carried out for 72 hours, giving 16 g/l of biomass containing ca. 55% w/w crude protein.  DOI:10.1002/elsc.201400213. A faster process, taking place under aerobic conditions in stirred-tank reactors, is now covered by the patent WO20191/21697A1 (2019, https://patents.google.com/patent/WO2019121697A1/en). However, the cultivation of filamentous fungi in bioreactors can be troublesome due to the tendency to entangle with the inner parts, such as baffles and impellers, leading to suboptimal mass and energy transfer rates. (https://doi.org/10.1002/elsc.201400213). Considering all these aspects, D. hansenii yeast appears to have a good cost / benefit ratio as protein producer. We really appreciated the Reviewer’s suggestions, so we decided to add some information about protein-rich products by other microorganisms into the text.

very active research field

 This strain is also able to degrade phytic acid by a cell-bound phytase activity.

Phytase activity  was also tested. Phytases are enzymes that catalyse the release of phosphate from phytic  acid and are commonly exploited as additive to reduce phytic acid content in feed/food.

 very interesting aspect

 it would be possible to produce natural compounds and relevant enzymes by means of seawater-based fermentations

 please provide current industrial productions based on seawater fermentation

 coupled with a wide range  of cheap carbon and nitrogen sources obtained from wastes.

 please provide current industrial productions based on seawater fermentation coupled with a wide range of cheap carbon and nitrogen sources obtained from wastes.

We could not find any information about processes already operated at industrial level, but we found several patents. A Chinese patent cover a novel seawater fermentation strain (Pseudomonas aeruginosa) generating a biosurfactant. In particular, fermentation with natural seawater as fermentation water and waste edible grease as a single carbon source is carried out to produce the biosurfactant rhamnolipid. (Patent n. CN104087525A).

Also the Actinobacteria Salinospora ATCC PTA-250 is already industrially used for the production of proteins, secondary metabolites and other biomolecules in seawater based media (Patent n. 20080070273).

In addition, several works reported the use of seawater for enzymatic hydrolysis stage (Grande at al., 2012) and pretreatment of lignocellulosic materials (vom Stein et al., 2012) and in fermentative processes using marine yeasts (Senthilraja et al., 2011; Lin et al. 2011; Gonçalves et al., 2015).

The combination of seawater with waste-derived C sources has been investigated for the production of bioplastic PHB by Halomonas campaniensis LS21 (Yue et al., 2014), for succinic acid production by Actinobacillus succinogenes using only wheat-derived medium and seawater (Lin et al. 2011) and for microbial oils production by Rhodotorula mucilaginosa using seawater instead of pure water with crude glycerol and yeast extract as C and N sources (Yen et al. 2016).

For protein production, the best results in seawater were obtained at lab scale by Cryptococcus aureus and its mutant Z114 (41% and 65%, respectively) (Zhang et al. 2009) and by Y. lipolytica SWJ-1b isolated from the marine fish gut and its mutant carrying the inulinase, contained from 47.6% to 53.7% (w/w) crude protein (Cui et al., 2011), values comparable to D.hansenii Mo40 level.

More recently, the potential of an integrated marine biorefinery system for industrial production has been widley investigated by Zaky, 2021 (https://doi.org/10.3390/pr9101841).

 or even coupled with a wide range of cheap carbon and nitrogen sources obtained from wastes, without seawater.

please enhance the quality of the captions

example

Figure 1. Growth of D. hansenii Mo40 strain under different conditions. SS indicates presence of sea salts. Blu: MMP pH  4.5, red: MMPSS pH 4.5, green: MMP pH 6, purple: MMPSS pH 6.

 Blue not Blu

all details should be provided in the caption so the reader is able to understand the figure without going throughout the text

DONE

 IMSS (simil-Industrial Medium): glucose 33 g/ L, xylose 16 g/ L, sea salt 40 g / L, yeast extract 2 g/ L, corn steep solid 5 g/ L, urea 2 g/ L, KH2PO4 3 g / L, H2SO4 2 mL/ L, CaCl2 0.1 g/ L, NaCl 0.1 g/ L.

 quite expensive medium with yeast extract…

We thank the Reviewer for the observation, however the use of yeast extract is often reported because advantageous in terms of costs/benefits, enhancing the yeast growth and reducing the fermentation time. In our study the industrial-like media was implemented by adding corn steep, a cheaper source widely present in industrial media formulation, nevertheless we will evaluate the total substitution of yeast extract with corn steep or others as malt extract or soy extract.

 comparison with MARINE BROTH contains all the nutrients necessary to cultivate the majority of marine bacteria, lacking the agar usual solidifier component. Our Marine Broth is prepared according to ZoBell, containing almost double the mineral content of sea water. The high salt content helps to simulate sea water.

or ZoBell??

ZoBell is a medium recommended for cultivation, isolation and enumeration of heterotrophic marine bacteria but the high content of peptone and other single minerals makes it not suitable for industrial purposes. In addition, the recommended final pH of ZoBell is 7.6, but low pH is preferred for industrial processes, to avoid contaminations.

In our work, the industrial-like media containing sea salts contains cheaper sources of nutrients, as urea and corn steep instead of ammonium nitrate.

Reviewer 4 Report

In this manuscript, Donzella and co-workers present an interesting study with high applicative potential regarding the use of a marine strain of Debaryomyces hansenii (Mo40) as a salt-tolerant and adaptable strain in applications suitable for circular economy. This study is even more relevant as D. hansenii has been included in QPS EFSA list (Quality Presumption as Safe by European Food Safety Autority).

The manuscript is very well written, presenting novel and sound data. There are issues that the authors need to address before the manuscript is considered for publication:

(1) To mimic the use of sea water instead of fresh water, the authors added 40 g/L of commercial sea salts to the synthetic media; it would be interesting to see the influence of different concentrations of sea salts on D. hansenii growth and propensity to produce biomass.

(2) The authors should explain the rationale behind using the “Lapeña” medium as one of the media tested.

(3) The authors found that the activity of D. hansenii (Mo40) phytase was higher at 60oC than at 37oC. This is a very interesting finding and the authors are encouraged to provide an explanation for this difference. 

Author Response

In this manuscript, Donzella and co-workers present an interesting study with high applicative potential regarding the use of a marine strain of Debaryomyces hansenii (Mo40) as a salt-tolerant and adaptable strain in applications suitable for circular economy. This study is even more relevant as D. hansenii has been included in QPS EFSA list (Quality Presumption as Safe by European Food Safety Autority).

The manuscript is very well written, presenting novel and sound data. There are issues that the authors need to address before the manuscript is considered for publication:

We appreciate the Reviewer’s positive feedback. The text has been checked and errors corrected.

  • To mimic the use of sea water instead of fresh water, the authors added 40 g/L of commercial sea salts to the synthetic media; it would be interesting to see the influence of different concentrations of sea salts on  hanseniigrowth and propensity to produce biomass.

We investigated the tolerance to different salt concentrations in a previous work (Capusoni et al. 2019), where a screening on medium containing dierent NaCl concentrations showed that among yeasts species isolated from animal fauna living at deep-sea hydrothermal vents (Burgaud et al., 2010), D. hansenii was the most halotolerant, being able to grow up to 2M NaCl (12%w/v). In this condition we found a decrease in growth rate and glucose consumption rate, while the presence of sea salts did not affect the growth of Mo40 strain.

Aiming to develop an industrial process using sea-water, we decided to investigate the performance on media containing the real concentration of sea salts (among 4%).

  • The authors should explain the rationale behind using the “Lapeña” medium as one of the media tested.

When we evaluated the protein content obtained in the industrial-like medium, we found a concentration of 23% of dry weight. This level of protein was lower in comparison to values reported by Lapeña, even though the amino acid composition was very similar. Being the medium reported by Lapeña and co-workers (glucose 20 g/ L, peptone 30 g/ L, yeast extract 20 g/ L) richer than the industrial-like medium, in order to understand if the lower protein content was due to the cultivation medium used, cells were cultivated under the same medium. We found that the protein content increased to 45%, indicating that the medium composition deeply affects the protein content of the yeast biomass.

  • The authors found that the activity of  hansenii(Mo40) phytase was higher at 60oC than at 37oC. This is a very interesting finding and the authors are encouraged to provide an explanation for this difference. 

Phytases usually exhibit high activity at temperatures ranging between 45 and 60°C (Vats & Banerjee, 2004). The maximal activity of three commercial microbial phytase (Aspergillus oryzae, A. niger, and Saccharomyces cerevisae) was found at high temperature values (> 41°C), corresponding to  the  ideal  physiological  conditions  of  broilers,  which  would  theoretically  allow  high  hydrolysis  rate  of  the  phytate contained in the feed (Naves at al., 2012).

Round 2

Reviewer 2 Report

I am happy for this to be published

Reviewer 3 Report

Revision conducted by authors is fine, from my side